# Reconstructing smoking history through dental cementum analysis - a preliminary investigation on modern and archaeological teeth

Valentina Perrone[1,2]*, Anna M Davies-Barrett[2], Mario Migliario[3],
Patrick Randolph-Quinney[4,5], Sarah A. Inskip[2], Edward C. Schwalbe[1]*

**1** Department of Applied Sciences, Faculty of Health and Life Sciences, Northumbria University, Newcastle upon Tyne, United Kingdom, **2** School of Archaeology and Ancient History, University of Leicester, University Road, Leicester, United Kingdom, **3** Department of Translational Medicine, University of Eastern Piedmont, Novara, Italy, **4** Department of Archaeology, Campus Gotland, Uppsala University, Sweden, **5** Department of Human Anatomy and Physiology, Faculty of Health Sciences, University of Johannesburg, Johannesburg, South Africa

\* valentina.perrone@northumbria.ac.uk; vp217@leicester.ac.uk (VP); ed.schwalbe@northumbria.ac.uk (ES)

## Abstract

Acellular extrinsic fibre cementum (AEFC) has been widely utilised in cementochronology to estimate age at death, seasonality, and for life-history reconstruction. Smoking has been commonplace in the UK since the 17th century and is known to compromise oral health and to modulate physiological processes. This study aimed to investigate whether AEFC analysis could identify smoking activity in both modern and archaeological populations. A modern sample (70 teeth from 46 donors) with known age, sex, and smoking status was compared with an archaeological sample (18 teeth from 18 individuals), dating from the 18th/19th centuries in Coventry, UK, whose biographical information was recorded from coffin plates where available. Smoking status for the archaeological individuals was inferred from pipe notches and dental staining. AEFC analysis that was blinded to smoking status measured increment count, overall width and the presence of irregularities within the cementum microstructure in both samples. Results demonstrated that the AEFC width was significantly lower (p = 0.008) in current smokers compared to ex-smokers. Additionally, individuals with a history of smoking were significantly more likely to display disrupted incremental patterns within their AEFC (p < 0.001). This research suggests an association between smoking and periodontal ligament health, which influences AEFC formation and shows that the AEFC provides a record of smoking-related oral health damage. This research expands the potential applications of cementochronology to forensic and archaeological investigations for life history reconstruction.

**Data availability statement:** All relevant data are within the article and its supporting information files.

**Funding:** SI reports funding from UK Research and Innovation – Future Leaders Fellowships grant MR/T022302/1. The funder had no role in study design, data collection and analysis, decision to publish, or preparation of the manuscript.

**Competing interests:** The authors have declared that no competing interests exist.

## Introduction

Teeth have proved to be useful for biological anthropology. Their macro- and micro-scopic analysis has revealed critical information on diet and migratory movements [1–3], age-at-death [4–6] and can contribute towards life-history reconstruction [7–11].

Teeth consist of three main hard tissues: enamel, dentine and cementum. Unlike the first two, dental cementum – and more specifically, the acellular extrinsic fibre cementum (AEFC) – continually grows on the coronal third of the dental root in alternating dark and light increments that are deposited until tooth loss or death of the individual. The continuity of cemental deposition is connected to its main role of anchoring the teeth in their sockets, withstanding masticatory and para-masticatory stresses. Like bones and other dental tissues, dental cementum also shows birefringent properties when observed in polarised light.

The pattern of cemental deposition is referred to under numerous names (*e.g.*, tooth cementum annulation or TCA [12]) and is defined as the seasonal formation of alternating light increments and dark increments, the latter clinically known as lines of Salter [13]. In the Northern hemisphere, light increments form between April and September, and the lines of Salter form between September and April (*i.e.*, respectively summer and winter season) [14]. Due to this seasonal phenomenon, each annulus (*i.e.*, a pair of light and dark increments) corresponds to one chronological year. By adding the total number of annuli to the tooth-specific age at eruption, the age of the individual can be calculated [15]. Stott et al. [16] were the first to use human dental cementum to estimate age in adults, effectively starting what is now known as cementochronology [4,17]. Methods commonly adopted in biological anthropology for assessment of age in adults rely on the degree of age-related degeneration of specific skeletal areas, such as the pelvis or sternal rib ends [18–20]. In comparison to these techniques, one advantage of cementochronology is that it is assessed on a tissue that grows throughout life and does not remodel, and on increments that seem to be internally regulated by a "circannual clock". In fact, the biological mechanisms of deposition of the AEFC are not well understood. Previously, the optical difference of the increments was thought to be due to a slower and faster rate of deposition of the tissue respectively occurring in the winter and summer season, which consequently resulted in a more organised and min-eralised layer in the winter (dark increment) and a less organised and mineralised layer in the summer (light-coloured increment) [21]. Contrary to this, more recent studies have now shown through environmental scanning electron microscopy and Raman spectroscopy imaging that light increments have, instead, a higher mineral content, effectively re-opening the debate on the biological mechanisms behind the incremental pattern of AEFC [22–25].

As well as being useful for estimating age and season at death, cementochronology has been shown to be useful for assessing and dating the occurrence of significant life events, such as pregnancies, weaning, skeletal traumas or diseases (especially if related to calcium metabolism) [7–10]. Specifically, disruptions in deposition have been correlated to the observation of considerably wider white

increments in pregnancy [10,23], skeletal traumas and renal disorders [10]. In Kagerer and Grupe [10], variations in increment thickness were also observed for the lines of Salter and, in one specific case, were associated with the prolapse of two intervertebral discs followed by injuries to the vertebrae. In another report [8], a number of physiological stressors (not only pregnancy, but also menopause, systemic illness and relocation) were identified on the AEFC by the occurrence of changes in the birefringent properties of cementum. The onset of puberty was also identified using cementochronology in both females and males [8,23]. In these studies, it was reported that variations in sex hormones were important regulators of cementum deposition.

Smoking is known to have a systemic impact on the body's physiology, leading to a cascade of effects that includes reduction in calcium intestinal absorption [26], alterations in vitamin D metabolism [26] and alterations to the thyroid, parathyroid, adrenal, pituitary glands and sex hormones [27]. Numerous studies [28–31] have also highlighted the correlation between smoking, periodontitis and tooth loss, suggesting that smoking activity could potentially be detected within the AEFC. However, to the authors' knowledge, no investigation has previously examined whether tobacco use can be predicted from analysis of AEFC. Since smoking can directly affect bone metabolism and oral health, the aim of this study was to assess whether tobacco consumption alters the dental cementum and its incremental growth. The hypothesis was that intense smoking activity might affect cementum increment formation and that periods of tobacco usage could be dated with cementochronology.

To test this, we analysed teeth from modern individuals with known smoking histories, and archaeological samples where it was possible to assess smoking status. As tobacco has one of the longest global histories of consumption as an intoxicant, the inclusion of archaeological smokers allowed us to explore how different ways and types of tobacco consumption might affect the AEFC in comparison to modern cigarettes. The pattern of cemental deposition of smokers and ex-smokers was also visually and statistically compared to that of non-smokers in order to identify irregularities that were potentially connected to smoking activity. Similar to the abovementioned studies addressing pathologies and life-history reconstruction, the cemental annulation of smokers and ex-smokers was expected to present some form of variation or disruption in the thickness and/or regularity of the increments.

The interest in understanding the impact of smoking on the AEFC is manifold, as it has clear application to the fields of dentistry, forensics and archaeology respectively by a) addressing potential factors altering the biology of the tissue and contributing to the currently ongoing research on periodontal regeneration; b) highlighting potential limitations in the application of cementochronology for age estimation and by providing a further identification tool when using dental cementum for life history reconstruction; and c) by providing archaeological and historical research with bio-anthropological evidence on the spread of tobacco in a given population.

## Materials & methods

A sample consisting of 88 human teeth (70 modern and 18 archaeological), representing a total of 64 individuals (46 modern and 18 archaeological), was selected for the analysis. Additional samples were collected for this study, but later excluded. Reasons for exclusion included dental roots being too pathologically compromised or damaged, and breakages occurring during sample preparation.

The modern subgroup was collected from living donors undergoing necessary dental treatment involving tooth extraction. Donors were all adult individuals (age range: 19–86 years), who were informed of the aims and objectives of the study and, if willing to participate, gave full informed consent on the use of their samples. Donors were asked to complete a questionnaire regarding their personal data (such as age, date of tooth extraction, medical and smoking record). The teeth donated for this study were collected with full informed consent from adult donors both at the University of Eastern Piedmont (UPO, Italy) and at Northumbria University (UK), where the study was conducted. Ethical approval to conduct this study was granted by the Northumbria University Research Ethics Committee on April 1st, 2021 (Ethical Ref. 29546). Sample collection was from April 2nd, 2021–31st March, 2022.

The total cohort (comprising both modern and archaeological samples) consisted of non-smokers (n = 33), ex-smokers (n = 17), smokers (n = 25), and for archaeological samples, included the additional categories potential smokers (n = 5), and indeterminate smoking status (n = 8) (Table S1). Through the questionnaire collected for the modern cohort, donors could identify themselves as non-smokers (*i.e.,* never smoked); ex-smokers (*i.e.,* used to smoke but had quit at the time of assessment); and smokers (*i.e.*, regularly smoking at the time of the assessment). The categories "potential smokers" and "indeterminate smoking status" were introduced to account for the uncertainty bound to archaeological specimens, whose smoking status relied on the interpretation of archaeological clues, such as dark dental staining and presence of pipe wear notches (alterations which are known to be associated with tobacco consumption (Fig 1), [33]). In this study, archaeological samples with clear evidence of pipe notches and stains were classified as "smokers"; samples with evidence potentially connected to smoking (*e.g.,* weak dental stains; small enamel alterations that could have been the onset of a pipe notch) were classified as "maybe" (*i.e.,* potential smokers); samples with unclear smoking evidence (*e.g.*, buried with a smoking pipe but no sign of smoking was identified on their dentition) were classified as "indet" (*i.e.,* indeterminate smoking status); samples with no evidence were classified as "non-smokers".

The archaeological samples were donated from the School of Archaeology and Ancient History at the University of Leicester (UK). These were sampled from skeletal remains excavated from Holy Trinity Church graveyard (on the site of St. Mary's Cathedral Church (site code: SMC99), Coventry, UK [32]), now preserved at the School of Archaeology and Ancient History, University of Leicester.

No permits were required for the analyses of these remains for the described study, which complied with all relevant regulations. These samples belonged to individuals dating to 1776–1890 and consisted of adults (n = 16) and sub-adults (n = 2) (age range: 2–65 years), for whom thirteen had recorded biological sex, age and date of death on their respective coffin plates. Five, however, did not have this information recorded so age was estimated to be between 35 and 50 years of age through observation of the pelvic region [18] by an experienced osteoarchaeologist.

In accordance with guidance for destructive analysis, each tooth was imaged digitally prior to sampling [34]. For the archaeological sample, teeth were re-imaged after the removal of dental calculus. Dental roots were embedded in a mixture of epoxy resin and hardener (ratio 5:2.5, EpoThin, Buheler©) and cast in cylindrical moulds of solid polytetrafluoroethylene (PTFE), which made the roots more stable and resistant for the sectioning procedure. The PTFE casts were internally covered in a thin layer of soft paraffin to ease the cast out once dried. After removal of the crown, the dental roots were positioned in the casts standing on their flat coronal surface and completely covered with the resin. The casts were then dried in a vacuum desiccator for 15–20 minutes and left to cure at room temperature for 24 hours. As the casts dried, the sample could be processed for sectioning. According to the main protocols for cementochronology [17,35–37], five transverse non-demineralised ground sections of 80 µm were produced from the coronal half of each tooth (where the

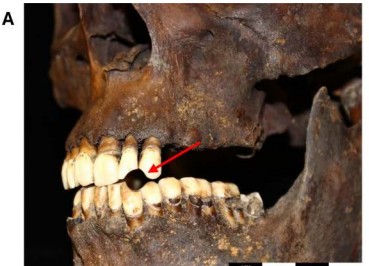 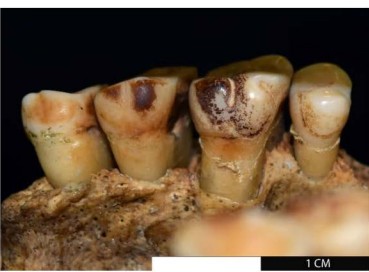

**Fig 1. Archaeological evidence of tobacco consumption.** Left: Example of pipe notch (arrowed) [SK134130, St James' Gardens Burial Ground, Euston, London (image reprinted from [33] under a CC BY license, with permission from Manchester University Press, original copyright 2024]. Right: Example of staining due to smoking [SK417, Holy Trinity Church, Coventry (image reprinted from [33], under a CC BY license, with permission from Manchester University Press, original copyright 2024)].

AEFC is normally found) by using a diamond wire saw (DWS.175, Diamond WireTech©) equipped with a micrometer. The thickness of the sections was set to a minimum of 80 µm to avoid unnecessary breakages during the sectioning procedure. This thickness is consistent with what is recommended for ideal visualisation of cementum annulations [17,35–37].

The same mixture of epoxy resin and hardener was then used to mount the sections on microscope slides which were covered with coverslips. This choice also followed the directions of previously successful studies [12] and the warnings that other mounting media (such as MMA) could hinder the correct visualisation of the increments [8].

Sections were analysed with a transmitted light microscope (Leica DM750©) fitted with an ICC50W camera module that permitted digital imaging of the sections as seen from the microscope. A 530nm retardation plate was also used to investigate the birefringent properties of the tissue. One region of interest (ROI) was selected for clarity and integrity of the AEFC per each section (n = 5) and recorded in a digital image. A total of five digital pictures were therefore recorded per each tooth. Digital images were saved in LAS X software at 20x and 40x magnification, for better visualisation of the cementum's features. Counts and measurements were carried out in ImageJ software (version Java 1.8.0_172). Increment counting was performed using the multipoint tool, which records the number of the increments while counting (therefore limiting potential mistakes). Measurements of the AEFC's width were taken at three equidistant points of each digital image. Counts of the increments and width measurements were then averaged by tooth in Excel. Assessment of age by cementochronology was performed according to previously published protocols [17,35–37] by adding the total pairs of the increments to the tooth-specific age at alveolar eruption (based on [38]). The term "eruption" refers more generally to the process of emergence of teeth in a functional (*i.e.*, occlusal) position within the oral cavity [39]. This process often is divided into three steps: alveolar emergence (tooth is over the alveolar bone); clinical emergence (tooth breaks through the gingiva); and functional occlusion (tooth is occluding with its opposing tooth) [40]. Since clinical emergence and functional occlusion could be more subjective to each individual [40], estimation of age through cementochronology was here based on the age at alveolar emergence.

Presence/absence of smoking damage was firstly assessed by visual observations of the sections and then further investigated by statistical analysis. The visual assessment consisted in recording whether structural variations and irregularities in the microstructure of the cementum (*e.g.*, alteration of the pattern; wider increments) were visible and could potentially be related to smoking activity. To further investigate the nature of this alteration, sections presenting damaged areas were also observed with the retardation plate. This allowed the comparison of the regular birefringent properties of the cemental increments with the areas of disruption. Once identified, the areas of smoking damage could also be aged through cementochronology.

Importantly, analyses of the sections were performed blind, without knowledge of smoking status; only during the final phase of the investigation was the smoking status of the donors disclosed.

### Statistical analyses

Statistical analysis was performed in R software (version 4.2.3) and visualised using ggplot2 (version 4.2.3). Association between the smoking habits and the smoking damage with the overall AEFC's width was assessed with Kruskal-Wallis (>2 groups) and Wilcoxon test (2 groups). The level of significance (p value) was set at 0.05. The association between true smoking status and predicted smoking status was tested using Fisher's Exact test. The accuracy of the predictions was further addressed by calculating the overall accuracy, sensitivity, specificity, positive predictive value (PPV) and negative predictive value (NPV). Since this was a pilot study, no correction for multiple testing was undertaken.

### Results

A visual assessment of the AEFC was carried out and compared between smokers, non-smokers and ex-smokers. The AEFC of smokers presented the usual cemental pattern (N = 16; Fig 2A), although it was of lower quality than that of non-smokers (Fig 2B). This manifested as less definition of the increments or, as in this case, in the presence of "dark

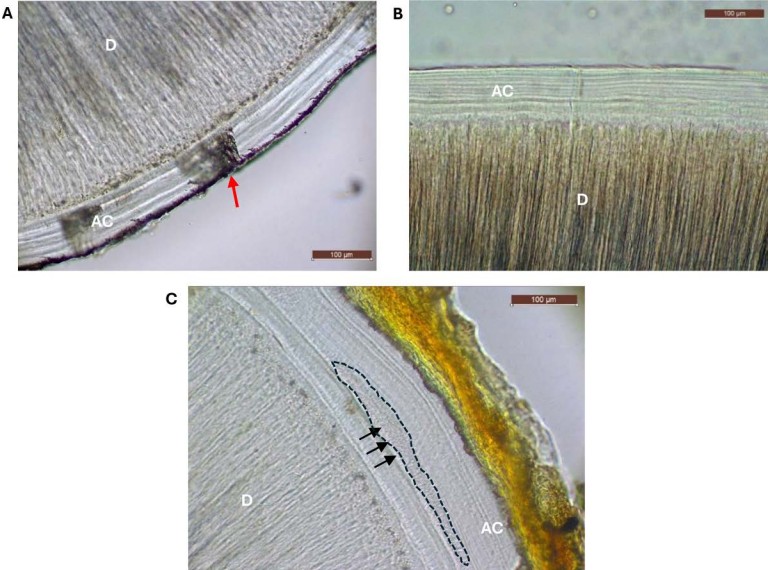

**Fig 2. Visual comparison of the AEFC in smoker, non-smoker and ex-smoker individuals.** Representative examples of acellular cementum (AC) in a 46-year-old smoker (A); in a 35-year-old non-smoker (B); and 58-year-old ex-smoker showing the smoking damage (black arrows and dashed line) (C). Teeth were affected by periodontal diseases (A, C) and one was impacted (B). [D=Dentin; Images taken with transmitted light microscopy at 20x, scale bar: 100µm].

areas" that spanned regions of the annulations (red arrow in Fig 2A). In contrast, the AEFC of ex-smokers presented strongly disrupted areas of tissue, in which the alternating cemental pattern was interrupted (black arrows in Fig 2C). These areas could be described as granular, structureless and "TV static"-like in appearance.

When observed with a retardation plate, areas of smoking damage appeared purple in colour despite orientating the sample in parallel or perpendicular to the axis of the plate (Fig 3). This showed that these areas did not have birefringent properties (unlike the rest of the tissue).

One of the teeth presenting smoking damage was from a donor who reported the exact age at which they started and ceased smoking. Cementochronology was performed to age the extent of the smoking damage observed on the section of this individual (Fig 2C). The total of annuli up to the start of the smoking-dependent damage was counted and added to the tooth-specific age at eruption (based on AlQathani [38]). To estimate the end of smoking-dependent damage, the total number of annuli counted from the external (most recent) side of the AEFC was subtracted from the known age of the individual. By doing so, the smoking damage observed on this individual was estimated to have occurred between age 22 and 41, which is congruent with the information collected by the donor, who reported to have started smoking at 28 years old and stopped at 38 years old (Table 1).

Data on sample VP_H_026 (shown in Fig 2C), indicating: type of tooth (FDI); Sex; Smoking status; Real age; Tooth-specific occlusion age; total count of the increments (IL Count); count of the increments up to the damage (smoking damage start) and from external border of the tissue (smoking damage end); prediction of age range at which the smoking damage occurred; and prediction of age of the individual.

Because the smoking damage was, in this case, close to the chronological time at which the donor reported to be a smoker, it was hypothesised to be caused by smoking activity and therefore, hereafter referred to as "smoking damage". As a result, each tooth of the cohort was examined for evidence of smoking damage and placed into one of three categories: Yes/ No/ indeterminate (indet). For donors, self-reported smoking status was classified as smokers, non-smokers and ex-smokers. For the archaeological samples, smoking status was estimated as described in the methods. The occurrence of smoking damage and smoking habits was then assessed in the context of chronological age and cementum width (Fig 4).

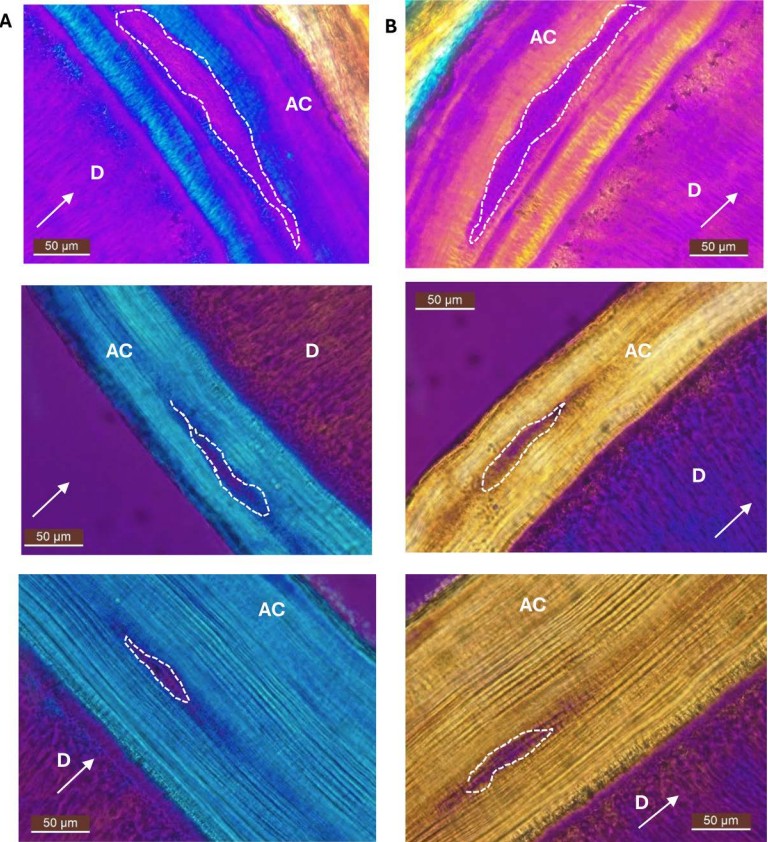

**Fig 3. Smoking-damaged areas show no birefringence.** Examples of the effects of the retardation plate on acellular cementum (AC) when perpendicular to the sample (column A) and parallel to the sample (column B). No birefringence property is shown on the smoking damage (dashed lines). Sample in top row is sample shown in Fig. 2C [D = Dentine; White arrow = orientation of the retardation plate; images taken with transmitted light microscopy at 40x, scale bar: 50µm].

**Table 1. Analyses on the smoking damage for prediction of individual's smoking timeline.** Data on sample VP_H_026 (shown in Fig 2C), indicating: type of tooth (FDI); Sex; Smoking status; Real age; Tooth-specific occlusion age; total count of the increments (IL Count); count of the increments up to the damage (smoking damage start) and from external border of the tissue (smoking damage end); prediction of age range at which the smoking damage occurred; and prediction of age of the individual.

| Sample | FDI | Sex | Smoking status | Real Age (years) | Occlusion Age (years) | IL Count | Smoking Damage start (IL) | Smoking Damage end (IL) | Age of smoking damage (years) | Predicted Age (years) |
|---|---|---|---|---|---|---|---|---|---|---|
| VP_H_026 | 1.3 | M | Ex Smoker (from age 28–38) | 58 | 12.5 | 41.70 | 10 | 17 | 22.5–41 | 54.20 |

There was a significant association between presence of smoking damage with smoking habits of the donors ($p < 0.001$, Fisher's Exact test; Table 2). Evidence of smoking damage was found in 8/24 (33%) of current smokers and in 7/10 (70%) of ex-smokers; this was rarely (1/32 (3%)) observed in non-smokers..

### Smoking damage is associated with smoking status ($p < 0.0001$, Fisher's Exact test)

The AEFC was also analysed in terms of overall width in relation to smoking habits of the donors and to the presence/absence of smoking damage. From this comparison, it was observed that in the modern subsample (Fig 5A), ex-smokers presented a thicker total AEFC than non-smokers and smokers ($p = 0.05$, Kruskal-Wallis). Interestingly, when comparing

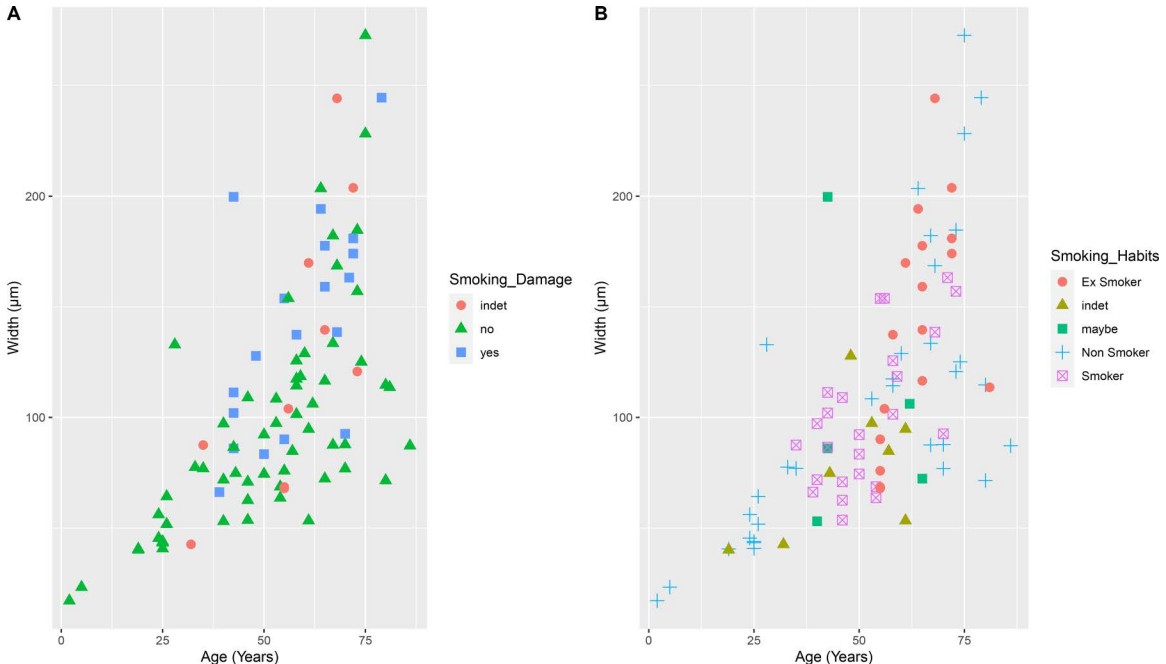

**Fig 4. Distribution of samples according to presence/absence of the smoking damage (A) and to the smoking habits of the donors (B).**

**Table 2. Association between occurrence of smoking damage by smoking habits. Smoking damage is associated with smoking status (p < 0.0001, Fisher's Exact test).**

| | Smoking Damage | |
|---|---|---|
| | **No** | **Yes** |
| **Non-Smoker** | 31 | 1 |
| **Ex Smoker** | 3 | 7 |
| **Smoker** | 16 | 8 |

the overall width with the presence/absence of smoking damage (Fig 5B), it was found that sections presenting smoking damage also showed a thicker AEFC width (p = 0.01, Kruskal-Wallis). There was a significant difference between smokers and ex-smokers (p = 0.008, Wilcoxon test) (Fig 5A), and between sections showing smoking damage and those not showing smoking damage (p = 0.004, Wilcoxon test) (Fig 5B). These differences were not as easily identifiable in the archaeological subsample (Fig 5C-D), and were not significant, possibly because of the smaller sample size (n = 18), and/or the number of teeth (n = 13) with indeterminate (*i.e.*, indet) or potential smoking evidence (*i.e.*, maybe). However, also in the archaeological subsample, sections presenting smoking damage were on average thicker than sections with no smoking damage (p = 0.02, Kruskal-Wallis; p = 0.006, Wilcoxon test; Fig 5D).

Overall, in the modern cohort, smoking damage was observed mostly on sections belonging to ex-smokers, rather than current smokers (Fig 6).

Finally, the power of prediction of smoking status through the observed presence of smoking damage was also calculated (Table 3). This showed that if smoking damage is observed, there is an 85–92% chance that the individual was either a smoker or ex-smoker (PPV); and that if smoking damage is not observed, there is a 96% chance that the individual was a non-smoker (specificity).

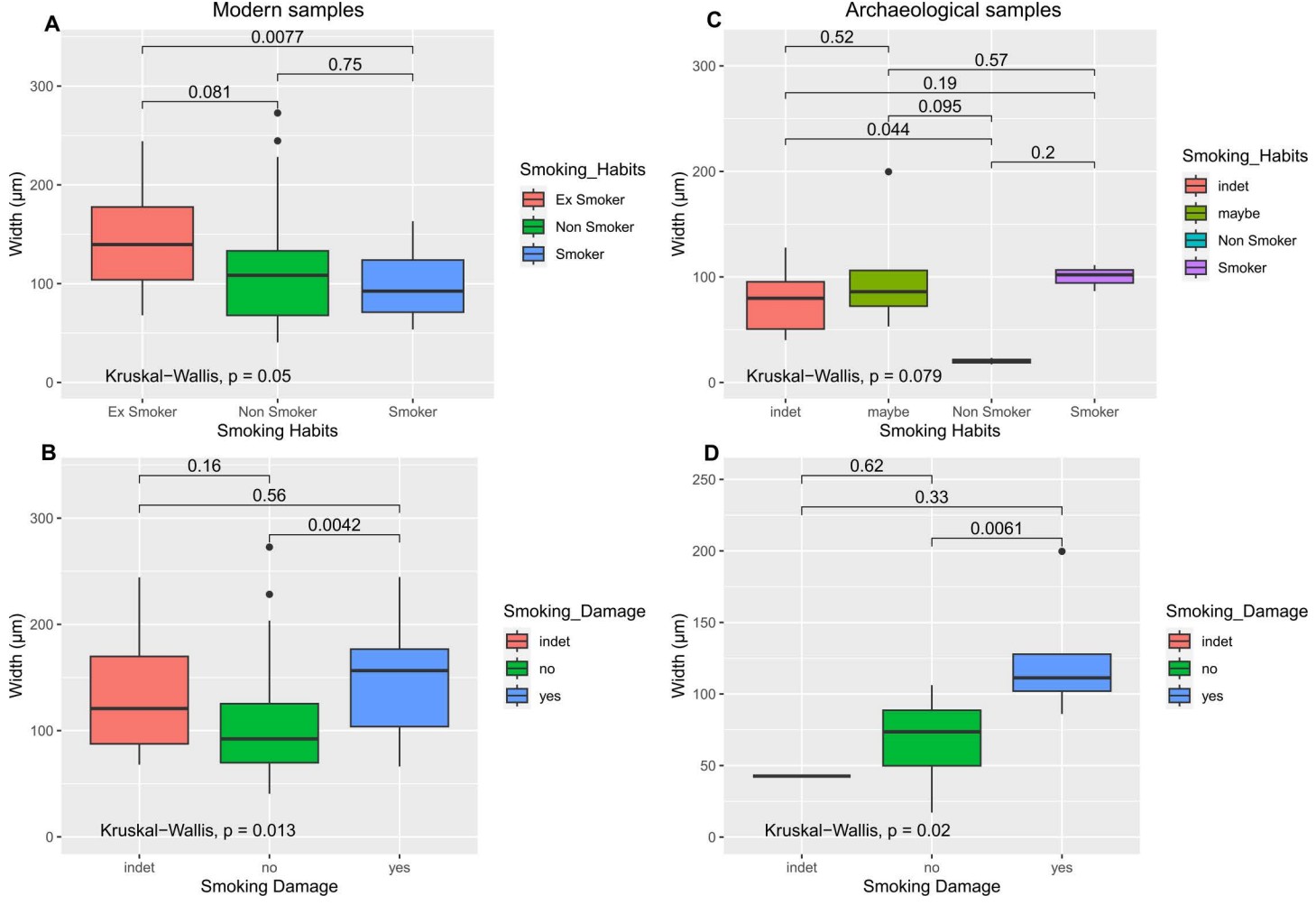

**Fig 5. Smoking status affects AEFC width.** Boxplots show AEFC width grouped by smoking habits for modern samples (A) and for archaeological samples (C). The relationship between presence/absence of smoking damage and AEFC width is shown for modern samples (B) and archaeological samples (D); further details in Table S2.

Calculations of the power of prediction of smoking status based on the observation of smoking damage, in terms of overall accuracy, sensitivity, specificity, positive predictive value (PPV) and negative predictive value (NPV).

A closer look into the archaeological subsample showed that, along with the observed pipe notches and dental staining associated with smoking, observation of smoking damage on the cementum could provide further evidence on the smoking habit of the individual (Table 4).

Comparison between the assessment of smoking status through archaeological examination and by presence of smoking damage on the AEFC. Indeterminate = "indet".

Fig 7 shows three archaeological samples in which smoking damage was most evident. Cementochronology was applied to attempt an estimation of the age at which smoking damage first occurred, resulting in age: 25.5 years old in sample ARC_009 (Fig 7A); 18 years old in sample ARC_015 (Fig 7B) and 24.5 years old in sample ARC_018 (Fig 7C).

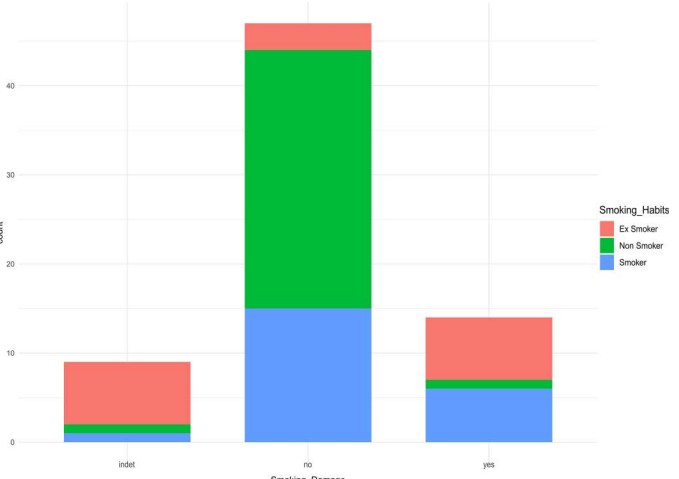

**Fig 6. Occurrence of smoking damage in non-smokers, ex-smokers and smokers.** Total count of the assessment for the smoking damage (indeterminate = "indet"; absent = "no"; present = "yes") in each smoking habit category.

**Table 3. Prediction of smoking habits. Calculations of the power of prediction of smoking status based on the observation of smoking damage, in terms of overall accuracy, sensitivity, specificity, positive predictive value (PPV) and negative predictive value (NPV).**

| Predicting for: | Overall accuracy | Sensitivity | Specificity | PPV | NPV |
|---|---|---|---|---|---|
| **All (Ex-Smokers & Smokers)** | 0.42 | 0.42 | 0.96 | 0.92 | 0.61 |
| **Smokers** | 0.35 | 0.28 | 0.96 | 0.85 | 0.65 |
| **Ex-Smokers** | 0.36 | 0.7 | 0.97 | 0.87 | 0.90 |

**Table 4. Evidence of smoking in archaeological samples. Comparison between the assessment of smoking status through archaeological examination and by presence of smoking damage on the AEFC. Indeterminate = "indet".**

| Sample ID | Repository Number | Sex | Smoking | |
|---|---|---|---|---|
| | | | Archaeological Record | Cementum |
| **VP_ARC_001** | **SMC 99_SK978** | M | indet | no |
| **VP_ARC_002** | **SMC 99_SK978** | M | indet | no |
| **VP_ARC_003** | **SMC 99_SK417** | F | possible staining | no |
| **VP_ARC_004** | **SMC 99_SK672** | F | indet | indet |
| **VP_ARC_005** | **SMC 99_SK50** | F | indet | no |
| **VP_ARC_006** | **SMC 99_SK516** | F | staining | no |
| **VP_ARC_007** | **SMC 99_SK1248** | M | indet | no |
| **VP_ARC_008** | **SMC 99_SK1041** | M | indet | no |
| **VP_ARC_009** | **SMC 99_SK808** | M | indet | yes |
| **VP_ARC_010** | **SMC 99_SK1195** | F | indet | no |
| **VP_ARC_011** | **SMC 99_SK1522** | M | staining | no |
| **VP_ARC_013** | **SMC 99_SK144** | M | indet | no |
| **VP_ARC_014** | **SMC 99_SK32** | NA | no | no |
| **VP_ARC_015** | **SMC 99_SK836** | M | staining | yes |
| **VP_ARC_016** | **SMC 99_SK1089** | M | staining | yes |
| **VP_ARC_017** | **SMC 99_SK400** | M | pipe notch | no |
| **VP_ARC_018** | **SMC 99_SK1225** | F | wear & staining | yes |
| **VP_ARC_020** | **SMC 99_SK85** | M | pipe notch & staining | yes |

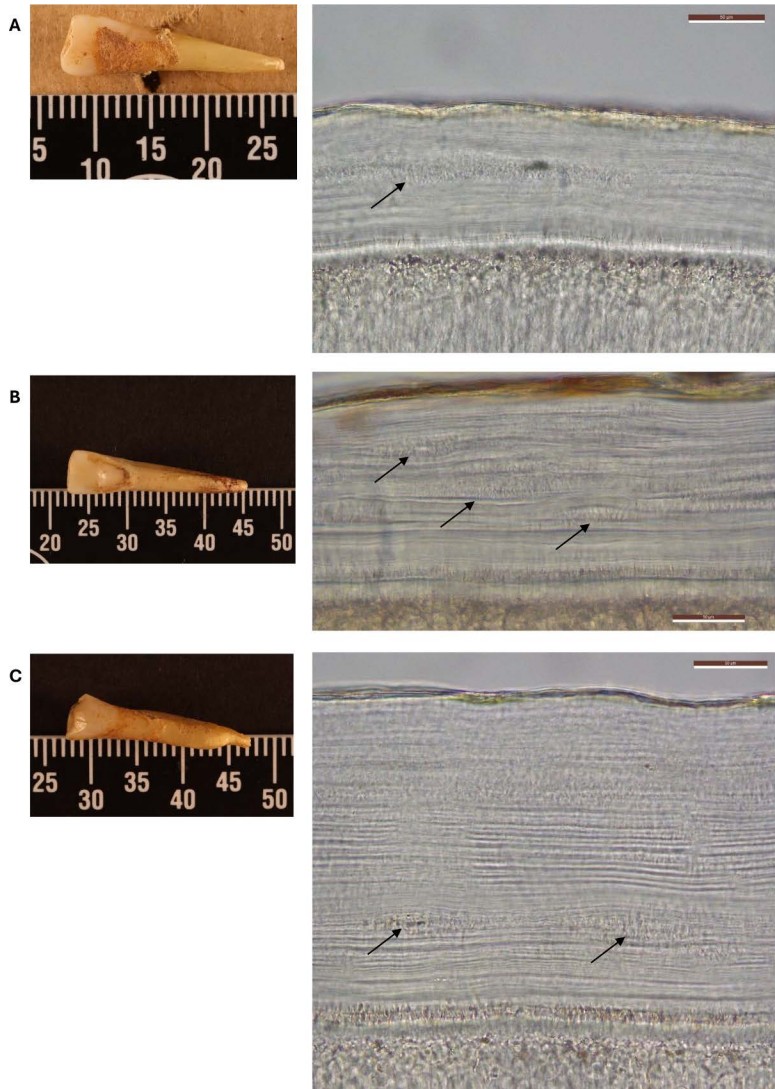

**Fig 7. Examples of the occurrence of the smoking damage in the archaeological sample (black arrows).** A) Sample VP_ARC_009; B) Sample VP_ARC_015; C) Sample VP_ARC_018 (transmitted light microscopy; images at 40x magnification; scale bar: 50μm; Image C, stitched in LAS X software).

## Discussion

The incremental non-remodelling nature of the AEFC allows for detection of illnesses and stressful events that, through cementochronology, can also be aged. Previously, events involving alterations of sex hormones and calcium metabolism have been positively identified on the AEFC [7,8,10,23]. In this study, the AEFC was investigated to determine whether smoking status could also be detected using the tissue, since it is known not only to carry a deeply disruptive action on the physiology of the body by altering its metabolism and hormone regulation, but also to have a direct effect on oral health.

The results of this preliminary investigation showed a significant association between smoking activity and the regular deposition of the AEFC, which seemed to be affected by smoking with different degrees of severity. These effects could be summarised overall with a) variations in the width of the tissue and b) presence of disrupted areas in which the typical

annulation pattern is not present (herein referred to as "smoking damage"). Interestingly, smoking damage was more often observed in ex-smokers, who also showed a thicker AEFC than non-smokers and smokers. Smoking damage appears as isolated areas of ground substance with an absence of increments. As the AEFC is known to be avascular and acellular, its non-remodelling nature could also explain a) why smoking damage is more often found in ex-smokers (*i.e.,* it becomes redefined by new incremental apposition once individuals stop smoking); and b) why the AEFC in ex-smokers is thicker than in other groups (smoking damage is not reabsorbed by the tissue, which instead resumes its regular deposition on top of it, resulting in a thicker cementum). These, however, are currently hypotheses that will need to be further investigated in future studies.

While the variations of the AEFC could be positively associated with the smoking status of modern donors, that association was not significant in the archaeological cohort (Fig 5C). This could be because the smoking status of these individuals was not certain but inferred, which led to most of the archaeological subsample having an indeterminate status (*i.e.,* "indet"; "maybe", n = 9). However, smoking damage observed in the AEFC of the archaeological samples was consistent with modern samples – smoking damage was significantly associated with a thicker cementum in both subgroups (Fig 5B–5D). When analysed with the retardation plate, smoking damage did not show any birefringent property, unlike the rest of the incremental pattern of the tissue, confirming that the increments do not form in these areas, and that they resume their regular pattern (along with their birefringent properties) once an individual quits smoking.

Unfortunately, the lack of consensus on the biology of the increments does not currently allow for a deeper understanding of what is being altered by smoking. According to Lieberman [41], for example, the two primary causes of the alternating pattern in cementum are variations in degree of mineralisation and in collagen orientation. While birefringence in cementum is normally associated with collagen orientation [41], Cool and colleagues [42] claimed, instead, that its birefringence and annulation are more closely related to the size and orientation of its crystals. More recent studies [22–25] found that the alternating pattern is indeed, as said by Lieberman [41], caused by both factors. In fact, they found that there is a difference in the degree of mineralisation, for which the bright increments show higher amounts of minerals (such as calcium, strontium, phosphorus, and zinc), but that the alternating increments also reflect peaks and drops in carbonate (hydroxy)apatite (cAp) as well as differences in collagen orientation.

The structureless and non-birefringent areas here associated with smoking suggest that smoking affects the AEFC both at its mineral and collagen level. Similar damage was also reported in two studies, where it was described as an area of high X-ray intensity, with no radial structure in the AEFC [43] and as a hypo-mineralised area of "amorphous appearance" in the cellular cementum [44]. Both studies looked at the effects of chronic inflammation of the PDL in acellular and cellular cementum and, in both cases, the damage observed was associated with the occurrence of severe periodontal diseases. Since a strong association between smoking and periodontal diseases [28–30,45,46] has been found, this would suggest that the occurrence of this specific area of damage could be directly caused by the development of periodontal diseases following intense smoking activity.

Amongst the localised side effects of smoking, there is, in fact, severe periodontitis, which is a major cause of tooth loss. Since periodontitis is a chronic inflammation of the PDL, it would frame smoking damage as more closely associated with the inflammatory-dependent damage already described by Yamamoto et al. and Simon et al. [43,44]. The AEFC is in fact, as the name suggests, acellular, and its growth is dependent on a layer of cells (cementoblasts) that lie on its external surface and are supported by the vascularised PDL [47]. Factors impacting the PDL would therefore impact the regular deposition of the cementum. Another study [48] also identified that estimates of age from teeth with untreated periodontitis and several degrees of alveolysis (resorption of the alveolar bone) were considerably underestimated with cementochronology, which is consistent with the idea that periodontitis, whether smoking-dependent or not, strongly affects cementum deposition.

The fact that the smoking damage also appeared on archaeological specimens with pipe notches and dental staining is interesting because, while the presence of smoking damage suggests that these individuals (similarly to modern

samples showing smoking damage) were ex-smokers rather than smokers, pipe notches and staining would tend to indicate a regular and persistent smoking activity (requiring about 4 years of regular pipe-smoking to leave a well-defined pipe notch, according to historical sources [49]). However, as shown in Fig 6, smoking damage is not always present in all ex-smokers and often its presence is not clear. This could indicate that the occurrence of smoking damage might be influenced by the quantity and ways (*i.e.*, smoking; snuff; chewing; etc.) in which tobacco was consumed rather than its frequency. Additionally, the presence of pipe notches on some of the archaeological specimens does not necessarily indicate a daily consumption and/or high quantities of tobacco but could have been caused by the use of a kind of clay pipe stem that was particularly abrasive [33]. To further complicate the matter, it is also unclear how pollution might have affected the evidence collected from the archaeological sample. In fact, recent studies regarding heavily polluted areas of South Korea and China [50,51] highlighted the relationship between air pollutants and the development of periodontitis. In particular, it was observed that the inhalation or ingestion of particulate matter (PM) (*i.e.*, ash and smoke) triggered an inflammatory response at both systemic (*i.e.*, metabolic, respiratory, cardiovascular diseases) and local levels (*i.e.*, oral tissues) [50]. Presence of heavy metals (such as iron, manganese and copper) in drinking water was also recently associated with the occurrence of periodontitis [52]. Since our archaeological samples belonged to individuals that lived during the Industrial Revolution (*circa* 1760–1850 [53]), it cannot be excluded that the behaviour of AEFC in relation to the presence of the smoking damage may have been further modulated by environmental pollution. The industrial revolution in England was a period characterised by high levels of pollution both in the air and drinking water, especially connected to the waste products of the growing industries related to coal and textiles [53].

Nonetheless, the simultaneous presence of dental staining, pipe notches and smoking damages on some of these samples suggests that individuals in 18th and 19th century Coventry (UK) were regularly smoking, but with varying quantities of tobacco. In some cases, more than one patch of smoking damage was observed at different points in the cementum (as in Fig 7B). Similarly, this could indicate periods of more and less intense smoking activity, potentially related to tobacco availability or their personal habits and methods of tobacco consumption. The principles of cementochronology were applied to age the smoking damage (similar to other studies for reconstruction of life-history events). Forensically, this provides a further aid to the identification of human remains, while archaeologically it gives a clue into the trends and habits of tobacco consumption in a historical context. In this study, three archaeological individuals were estimated to have started consuming tobacco between the age of 18 and 25 years old. This would be consistent with the findings observed in a study of 234 individuals from three different locations of 1700-1800s England (one of which, Coventry, UK was the same population of our archaeological sample) that, from the analysis of dental staining and pipe notches, suggested that individuals were already smoking during (and even before) their adolescence [33].

The analysis of pathological samples extracted from individuals that were not only smokers but also had other underlying tooth pathology might have limited our understanding of the impact of smoking on the AEFC. Similarly, the unknowns concerning the urban environment in which the archaeological populations lived as well as the ways and frequency of their tobacco consumption, also represent another limitation. Despite these limitations, the preliminary findings of this study offer a foundation for future studies into the biology of the AEFC and the application of cementochronology for age estimation and life history reconstruction. Future studies should aim to a) understand the biological alteration of the smoking damage to further explore the biology of the increments; b) clarify how and when the smoking damage occurred, by accurately recording the number of cigarettes smoked and investigating other ways of smoking (*e.g.*, vaping); c) study the smoking trends of past populations by selecting a larger cohort presenting archaeological evidence of smoking and the comparison with evidence from historical records regarding type of tobacco consumed, trade routes and rates of consumption.

## Conclusions

Overall, this investigation into the effects of smoking on the dental cementum shows how the time-keeping and non-remodelling nature of the AEFC can be useful when reconstructing life-history events of past and modern individuals, adding to personal lifestyles and to population trends. Smoking habits were here investigated and analysed with cementochronology to investigate whether teeth from smokers and ex-smokers could be distinguished from never smokers. Further questions were raised about the occurrence of the smoking damage: how the smoking damage changes according to quantity of tobacco consumed, and type of smoking (*e.g.*, vaping; snuff), and whether similar damage occurs in non-smokers that are affected by severe periodontitis. Future research could address these questions in modern populations, while, for archaeological specimens, a wider comparison between individuals living in rural and urban areas should be carried out, to better account for the potential effects of pollution on the AEFC.

This study also highlighted the level of physiological disruption that smoking causes to the body, and more specifically to the dental cementum. The smoking damage is not commonly addressed in the current literature and was only described in two studies published in the 1960s and 1980s that did not associate it with smoking but with high levels of inflammation of the PDL [43,44]. The occurrence of a similar kind of damage in ex-smokers would point at the strict interconnection between cemental increments and PDL, adding to the ongoing research directed at understanding the biology of AEFC increments.

## Supporting information

**Table S1. Study dataset.** Complete data and metadata of modern and archaeological samples. The table summarizes samples metadata regarding the time period (archaeological or modern); tooth type (following FDI identification system); dental pathologies; smoking habits; and real age of the samples (Real_Age and Real_Age_1 – the latter has been added for the archaeological samples, whose age was estimated and whose age range was averaged for the purpose of this study). The table also shows the outcome of the analyses carried out in this study (occurrence of the smoking damage; the total count of the increments counted (IL_Count); the measurement of cementum width (Width); and prediction of age (Predicted_Age)). "Occlusion_Age" refers to the standardised age at which teeth come into occlusion, according to AlQahtani et al., 2010 [38].
(XLSX)

**Table S2. Kruskal-Wallis rank sum test.** Summary of the chi-squared and p-values between cementum width and smoking habits and damage, in overall cohort, modern and archaeological subsamples.
(DOCX)

## Acknowledgments

The authors would like to thank Dr Sebastian Breitenbach for training on the use of the diamond wire saw, and Mr Philip Donnelly for manually producing the moulds and samples holder fitting the diamond wire saw at the department of Geography and Engineering, Northumbria University (UK). Finally, our thanks go to Dr Ralf Kist at Newcastle Dental School (UK) for their recommendations on this paper.

## Author contributions

**Conceptualization:** Valentina Perrone, Patrick Randolph-Quinney.

**Data curation:** Valentina Perrone, Sarah A Inskip.

**Formal analysis:** Valentina Perrone, Anna Davies-Barrett.

**Investigation:** Valentina Perrone, Anna Davies-Barrett, Patrick Randolph-Quinney.

**Methodology:** Valentina Perrone, Anna Davies-Barrett, Patrick Randolph-Quinney, Sarah A Inskip.

**Project administration:** Valentina Perrone, Patrick Randolph-Quinney, Sarah A Inskip, Edward C Schwalbe.

**Resources:** Mario Migliario, Sarah A Inskip.

**Supervision:** Patrick Randolph-Quinney, Edward C Schwalbe.

**Visualization:** Valentina Perrone, Edward C Schwalbe.

**Writing – original draft:** Valentina Perrone, Edward C Schwalbe.

**Writing – review & editing:** Valentina Perrone, Anna Davies-Barrett, Mario Migliario, Patrick Randolph-Quinney, Sarah A Inskip, Edward C Schwalbe.

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
