## [Decision Letter · Decision Letter 0]

25 Feb 2025

PONE-D-25-00796Reconstructing smoking history through dental cementum analysis - a preliminary investigation on modern and archaeological teeth.PLOS ONE

Dear Dr. Schwalbe,

Thank you for submitting your manuscript to PLOS ONE. After careful consideration, we feel that it has merit but does not fully meet PLOS ONE’s publication criteria as it currently stands. Therefore, we invite you to submit a revised version of the manuscript that addresses the points raised during the review process.

We look forward to receiving your revised manuscript.

Kind regards,

Francisco Wanderley Garcia de Paula-Silva, DDS, MSc, PhD

Academic Editor

PLOS ONE

2. In your manuscript, please provide additional information regarding the specimens used in your study. Ensure that you have reported human remain specimen numbers and complete repository information, including museum name and geographic location.

For more information on PLOS ONE's requirements for paleontology and archeology research, see https://journals.plos.org/plosone/s/submission-guidelines#loc-paleontology-and-archaeology-research .

 [Dr. Inskip reports funding from UK Research and Innovation – Future Leaders Fellowships grant MR/T022302/1]. 

5. We are unable to open your Figure file [Fig_1.eps to Fig_7.eps]. Please kindly revise as necessary and re-upload.

Reviewers' comments:

Reviewer's Responses to Questions

**Comments to the Author**

1. Is the manuscript technically sound, and do the data support the conclusions?

Reviewer #1: Yes

Reviewer #2: Yes

2. Has the statistical analysis been performed appropriately and rigorously? 

Reviewer #1: Yes

Reviewer #2: Yes

3. Have the authors made all data underlying the findings in their manuscript fully available?

Reviewer #1: Yes

Reviewer #2: Yes

4. Is the manuscript presented in an intelligible fashion and written in standard English?

Reviewer #1: Yes

Reviewer #2: Yes

5. Review Comments to the Author

Reviewer #1: Reconstructing the life-history of people who lived in the past is one of the most exciting fields in bioarchaeology. Often, such reconstructions are made using only paleoanthropological data, such as estimating smoking prevalence based on tooth wear and dental plague studies, supplemented by historical records. This study is notable not only because it uses a new method for estimating smoking effects on an organism by analyzing increments of dental cementum. It also allows for a comparison of the use of this method in different populations: a modern one with a known human smoking history and an archaeological one, where only indirect evidence is used.

The article is written in fluent language. The problem and the relevance of the research are well stated. The methodology is clearly described, and appropriate statistical methods are applied. The results are presented clearly, and the tables and figures are informative. The discussion covers not only possible interpretations of the results but also the limitations of this study and future directions.

There are only a few clarifying questions that need to be addressed:

1. How was it established that the damages found in the study were related precisely to the effects of smoking? The authors cite the coincidence between the actual smoking period reported by one of the respondents and the time at which the cementum lesions developed. Is this coincidence the only basis for the argument?

2. The authors discuss the possibility that other causes, such as air pollution and metabolic or infectious diseases, could have contributed to the damages found in the archaeological material. Identifying the specific cause of damage in skeletal material in archaeological populations is usually impossible. However, have the links between smoking and comorbidities or injuries in the modern population not been investigated? An analysis of these associations would help better interpret the results obtained. Moreover, the authors provide a questionnaire in Supplements with questions on malnutrition or trauma.

3. The questionnaire form provided in the Supplements implies that extensive additional information was collected, but only data on smoking habits were used in the survey. Table S1 also includes information on dental diseases, but these are barely addressed in the text, except to mention the link between smoking and periodontitis. Are there plans to use the other information collected, and how?

However, the above comments do not detract from the value of the paper. The paper is certainly worth publishing.

Reviewer #2: The study provided a preliminary investigation using a sound methodology that has the potential to be applied to other future studies. The statistic analysis was rigorous and all data underlying the findings described in their manuscript were fully available. The language used was appropriate for the audience and the topic. A few structural modifications are suggested to improve the overall presentation and some of the interpretations given to the results.

Reviewer #2: 

The paper represents a novel approach to the study of smoking status in past populations with the use of cementochronology. The authors provided an in-depth literature review on the topic including the answers that AEFC can provide to specific osteoarchaeological questions. The study provided a preliminary investigation using a sound methodology that has the potential to be applied to other future studies. The Discussion provided a good interpretation of the results. It also addressed the issues found with the archaeological sample and suggested areas of future potential development. Overall, the paper represents a great contribution to the field although some aspects of the Materials-Methods and Results could be explained and/or presented better. 

-The **Materials and Methods ** defined the archaeological samples in Lines 121-122 (Smokers, Potential smokers, Indet, Non-smoker) but the modern samples were defined in the **Results ** (smokers, non-smokers and ex-smokers). I have to admit that I had to read these sections more than once to understand what sample was being referred to and how. This is the reason I believe the Materials and Methods – and Results would be better presented had these terms been explained in one place (the methods). If the donors' self-reported status is listed/defined in the methods, then the results section could present the data according to the cementochronology. 

-Also, the term ‘smoking damage’ is defined in Line 238 but sometimes ‘damage’ was used interchangeably to mean ‘smoking damage’. I’d recommend sticking to this term instead of just calling it ‘damage’ because the word ‘damage’ was used in other sections of the paper to mean different things. So, to not create confusion I’d recommend you to state the full word ‘smoking damage’ to make a clear distinction from other meanings.  

**Discussion:**

-The discussion mentions that recent studies in South Korea and China highlighted the relationship between air pollutants and the development of periodontitis. The assumption that your archaeological samples belonged to individuals who lived during the Industrial Revolution implied that they would have been exposed to environmental pollution. Now, specifically to your archaeological sample from industrial Coventry, what kind of potential pollution would have been the individuals exposed to that could justify the behaviour of AEFC in relation to the presence of smoking damage?

Other minor comments:

-Figure legend for figure 1 needs to say what context number/skeleton number it is

-Line 244: although technically you can start a sentence with a number ‘8/24’ it doesn’t look appropriate for a formal paper to do it.

6. PLOS authors have the option to publish the peer review history of their article (what does this mean? ). If published, this will include your full peer review and any attached files.

**Do you want your identity to be public for this peer review?** For information about this choice, including consent withdrawal, please see our Privacy Policy .

Reviewer #1: No

Reviewer #2: No

---

## [Author Response · Author response to Decision Letter 1]

1 Apr 2025

Please see attached point-by-point response to reviewers and to the Academic editor. [Response to Reviewers.docx]

Reviewer 1

Reconstructing the life-history of people who lived in the past is one of the most exciting fields in bioarchaeology. Often, such reconstructions are made using only paleoanthropological data, such as estimating smoking prevalence based on tooth wear and dental plague studies, supplemented by historical records. This study is notable not only because it uses a new method for estimating smoking effects on an organism by analyzing increments of dental cementum. It also allows for a comparison of the use of this method in different populations: a modern one with a known human smoking history and an archaeological one, where only indirect evidence is used.

The article is written in fluent language. The problem and the relevance of the research are well stated. The methodology is clearly described, and appropriate statistical methods are applied. The results are presented clearly, and the tables and figures are informative. The discussion covers not only possible interpretations of the results but also the limitations of this study and future directions.

We thank the reviewer for their positive feedback and for their support on this study. We are very glad that they have appreciated the value of this work and its contribution to the field of bioarchaeology.

1. How was it established that the damages found in the study were related precisely to the effects of smoking? The authors cite the coincidence between the actual smoking period reported by one of the respondents and the time at which the cementum lesions developed. Is this coincidence the only basis for the argument?

We thank the reviewer for raising this point. This coincidence is not the only basis for the argument, but rather is what drew our attention to the occurrence of smoking damage in smokers and ex-smokers. Through this more in-depth investigation, we demonstrate a significant association between consumption of tobacco and presence of damage in the wider cohort. This was also supported by the fact that the presence of the damage was also highly predictive of the smoking habit of the individuals.

Unfortunately, information on the precise time period between commencing smoking and its cessation was only available for one donor. In follow up studies, we are currently working on a new cohort of modern samples, where smoking timelines are more clearly defined.

2. The authors discuss the possibility that other causes, such as air pollution and metabolic or infectious diseases, could have contributed to the damages found in the archaeological material. Identifying the specific cause of damage in skeletal material in archaeological populations is usually impossible. However, have the links between smoking and comorbidities or injuries in the modern population not been investigated? An analysis of these associations would help better interpret the results obtained. Moreover, the authors provide a questionnaire in Supplements with questions on malnutrition or trauma.

We thank the reviewer for highlighting such an important point and agree that a better insight into modern comorbidities and injuries would help elucidate the complex interplay of factors that may engender damage to the AEFC.

This is why the questionnaire included questions related to presence of pathologies and conditions affecting the metabolism of vitamin C and D, for example. The section of the questionnaire regarding presence of pathologies, smoking, etc. was left at the donor’s discretion (see “Part 2- Optional”). As a result, not all donors completed the second part of the questionnaire, leaving us with data that was not adequately powered to make statistical inferences. As mentioned above, we are now working towards rectifying the gaps we identified during this pilot study in future works.

3. The questionnaire form provided in the Supplements implies that extensive additional information was collected, but only data on smoking habits were used in the survey. Table S1 also includes information on dental diseases, but these are barely addressed in the text, except to mention the link between smoking and periodontitis. Are there plans to use the other information collected, and how?

We thank the reviewer for this question. Please refer to the answer for A.2 which discusses the limited nature of additional information.

While the link between periodontal diseases and cementum alteration was more clearly noticeable (a point already addressed and highlighted in the literature on cementochronology - as in Broucker, Colard, Penel et al., 2016 - https://doi.org/10.1016/j.ijpp.2015.09.004), once again, the small number of samples with specific dental pathologies (i.e. caries and impaction) did not allow statistical analysis. The information collected on these other dental pathologies were therefore excluded from this pilot, but will be integrated in following studies.

However, the above comments do not detract from the value of the paper. The paper is certainly worth publishing.

We thank the reviewer once more for their support in our work, and for giving us the opportunity to further clarify the rationale behind our study.

Reviewer 2

The study provided a preliminary investigation using a sound methodology that has the potential to be applied to other future studies. The statistic analysis was rigorous and all data underlying the findings described in their manuscript were fully available. The language used was appropriate for the audience and the topic. A few structural modifications are suggested to improve the overall presentation and some of the interpretations given to the results.

The paper represents a novel approach to the study of smoking status in past populations with the use of cementochronology. The authors provided an in-depth literature review on the topic including the answers that AEFC can provide to specific osteoarchaeological questions. The study provided a preliminary investigation using a sound methodology that has the potential to be applied to other future studies. The Discussion provided a good interpretation of the results. It also addressed the issues found with the archaeological sample and suggested areas of future potential development. Overall, the paper represents a great contribution to the field although some aspects of the Materials-Methods and Results could be explained and/or presented better.

We thank the reviewer for their positive feedback on our manuscript and welcome the suggested improvements.

1. The Materials and Methods defined the archaeological samples in Lines 121-122 (Smokers, Potential smokers, Indet, Non-smoker) but the modern samples were defined in the Results (smokers, non-smokers and ex-smokers). I have to admit that I had to read these sections more than once to understand what sample was being referred to and how. This is the reason I believe the Materials and Methods – and Results would be better presented had these terms been explained in one place (the methods). If the donors' self-reported status is listed/defined in the methods, then the results section could present the data according to the cementochronology.

We thank you the reviewer and agree that the definitions could benefit from a more unified definition with greater clarity. We have provided further definition on the terms in the Methods section - please find our changes in lines 131-146.

“The total cohort (comprising both modern and archaeological samples) consisted of non-smokers (n=33), ex-smokers (n=17), smokers (n=25), and for archaeological samples, included the additional categories potential smokers (n=5), and indeterminate smoking status (n=8) (Table S1). Through the questionnaire collected for the modern cohort, donors could identify themselves as non-smokers (i.e. never smoked); ex-smokers (i.e. used to smoke but had quit at the time of assessment); and smokers (i.e., regularly smoking at the time of the assessment). The categories “potential smokers” and “indeterminate smoking status” were introduced to account for the uncertainty bound to archaeological specimens, whose smoking status relied on the interpretation of archaeological clues, such as dark dental staining and presence of pipe wear notches (alterations which are known to be associated with tobacco consumption (Fig 1), [33]). In this study, archaeological samples with clear evidence of pipe notches and stains were classified as “smokers”; samples with evidence potentially connected to smoking (e.g., weak dental stains; small enamel alterations that could have been the onset of a pipe notch) were classified as “maybe” (i.e., potential smokers); samples with unclear smoking evidence (e.g., buried with a smoking pipe but no sign of smoking was identified on their dentition) were classified as “indet” (i.e., indeterminate smoking status); samples with no evidence were classified as “non-smokers”.”

We hope this is now more clearly presented and satisfies the reviewer’s recommendations.

2. Also, the term ‘smoking damage’ is defined in Line 238 but sometimes ‘damage’ was used interchangeably to mean ‘smoking damage’. I’d recommend sticking to this term instead of just calling it ‘damage’ because the word ‘damage’ was used in other sections of the paper to mean different things. So, to not create confusion I’d recommend you to state the full word ‘smoking damage’ to make a clear distinction from other meanings.

We thank you the reviewer for their suggestions. We have apported the recommended change throughout the manuscript (please see the revised manuscript with tracked changes).

3. The discussion mentions that recent studies in South Korea and China highlighted the relationship between air pollutants and the development of periodontitis. The assumption that your archaeological samples belonged to individuals who lived during the Industrial Revolution implied that they would have been exposed to environmental pollution. Now, specifically to your archaeological sample from industrial Coventry, what kind of potential pollution would have been the individuals exposed to that could justify the behaviour of AEFC in relation to the presence of smoking damage?

We thank the reviewer for raising a such an interesting point. We have hypothesised that the occurrence/visibility of the damage might be accentuated by the quantity of tobacco consumed by the individual. This hypothesis is being investigated in a new study we are currently finalising, which has demonstrated that heavy smokers are more likely to present damage compared to medium or occasional smokers.

Following this logic, we hypothesized that a greater exposure to environmental pollution might have worsened the dental cementum health in archaeological individuals (in the discussion, we have addressed this in lines 404-409). From historical sources, we know that, during the Industrial Revolution, individuals were exposed daily to pollutants in the water they were drinking and in the air they were breathing both outside (e.g., factories) and inside their houses (i.e., open fires for cooking/heating), smog produced by factories, of which there were many in Coventry. These would have included pollutants such as lead and carbon monoxide, that are also released and inhaled while smoking. This might have had an impact on the dental cementum, and in what we have observed in this study. Environmental pollution would have affected all individuals, regardless of their smoking habits. However, exposure to environmental pollution only would have probably not been enough to cause the smoking damage; whereas it could have added to the ongoing oral inflammation that smokers were subjected to due to their smoking habit.

However, since this was a pilot, we did not have enough data to investigate whether this is indeed the case, therefore, we have mentioned it as a hypothesis. In future work, we are seeking to address this point by comparing archaeological and modern smokers living in urban areas and rural areas.

4. Figure legend for figure 1 needs to say what context number/skeleton number it is

We thank the reviewer for bringing this to our attention. Please find the recommended change at line 206-209.

“Fig 1. Archaeological evidence of tobacco consumption. Left: Example of pipe notch (arrowed) [SK134130, St James’ Gardens Burial Ground, Euston, London (image reprinted from [33])]. Right: Example of staining due to smoking [SK417, Holy Trinity Church, Coventry (image reprinted from [33])].”

5. Line 244: although technically you can start a sentence with a number ‘8/24’ it doesn’t look appropriate for a formal paper to do it

We thank the reviewer for bringing this to our attention. Please find the recommended change at lines 257-259.

“Evidence of smoking damage was found in 8/24 (33%) of current smokers and in 7/10 (70%) of ex-smokers; this was rarely (1/32 (3%)) observed in non-smokers.”

---

## [Decision Letter · Decision Letter 1]

16 Apr 2025

Reconstructing smoking history through dental cementum analysis - a preliminary investigation on modern and archaeological teeth.

PONE-D-25-00796R1

Dear Dr. Schwalbe,

We’re pleased to inform you that your manuscript has been judged scientifically suitable for publication and will be formally accepted for publication once it meets all outstanding technical requirements.

Kind regards,

Francisco Wanderley Garcia de Paula-Silva, DDS, MSc, PhD

Academic Editor

PLOS ONE

Additional Editor Comments (optional):

Reviewers' comments:

Reviewer's Responses to Questions

**Comments to the Author**

1. If the authors have adequately addressed your comments raised in a previous round of review and you feel that this manuscript is now acceptable for publication, you may indicate that here to bypass the “Comments to the Author” section, enter your conflict of interest statement in the “Confidential to Editor” section, and submit your "Accept" recommendation.

Reviewer #1: All comments have been addressed

2. Is the manuscript technically sound, and do the data support the conclusions?

Reviewer #1: Yes

3. Has the statistical analysis been performed appropriately and rigorously? 

Reviewer #1: Yes

4. Have the authors made all data underlying the findings in their manuscript fully available?

Reviewer #1: Yes

5. Is the manuscript presented in an intelligible fashion and written in standard English?

Reviewer #1: Yes

6. Review Comments to the Author

Reviewer #1: I thank the authors for their detailed comments on the questions raised. The comments have been very valuable for a better understanding of the results presented in the paper and clearly outlined the way forward for future research.

7. PLOS authors have the option to publish the peer review history of their article (what does this mean? ). If published, this will include your full peer review and any attached files.

**Do you want your identity to be public for this peer review?** For information about this choice, including consent withdrawal, please see our Privacy Policy .

Reviewer #1: No

---

## [Editor Report · Acceptance letter]

PONE-D-25-00796R1

PLOS ONE

Dear Dr. Schwalbe,

I'm pleased to inform you that your manuscript has been deemed suitable for publication in PLOS ONE. Congratulations! Your manuscript is now being handed over to our production team.

Kind regards,

on behalf of

Prof. Dr. Francisco Wanderley Garcia de Paula-Silva

Academic Editor

PLOS ONE